# The role of tropical upwelling in explaining discrepancies between recent modeled and observed lower stratospheric ozone trends

Sean M. Davis[1], Nicholas Davis[2], Robert W. Portmann[1], Eric Ray[1,3], Karen Rosenlof[1]

[1] NOAA Chemical Sciences Laboratory, Boulder, CO, USA

[2] Atmospheric Chemistry Observations and Modeling Laboratory, National Center for Atmospheric Research, Boulder, CO, USA

[3] Cooperative Institute for Research in Environmental Sciences, University of Colorado at Boulder, Boulder, CO, USA

*Correspondence to*: Sean Davis (sean.m.davis@noaa.gov)

**Abstract.** Several analyses of satellite-based ozone measurements have reported that lower stratospheric ozone has declined since the late 1990s. In contrast to this, lower stratospheric ozone was found to be increasing in specified dynamics (SD) simulations from the Whole Atmosphere Community Climate Model (WACCM-SD), despite the fact that these simulations are expected to represent the real-world dynamics and chemistry relevant to stratospheric ozone changes. This paper seeks to explain this specific model/observational discrepancy and to more generally examine the relationship between tropical lower stratospheric upwelling and lower stratospheric ozone. This work shows that, in general, the standard configuration of WACCM-SD fails to reproduce the tropical upwelling changes present in its input reanalysis fields. Over the period 1998 to 2016, WACCM-SD has a spurious negative upwelling trend that induces a positive near-global lower stratospheric column ozone trend and accounts for much of the apparent discrepancy between modeled and observed ozone trends. Using a suite of SD simulations with alternative nudging configurations, it is shown that short-term (~2 decade) lower stratospheric ozone trends scale linearly with short-term trends in tropical lower stratospheric upwelling near 85 hPa. However, none of the simulations fully capture the recent ozone decline, and the ozone/upwelling scaling in the WACCM simulations suggests that a large short-term upwelling trend (~6% decade$^{-1}$) would be needed to explain the observed satellite trends. The strong relationship between ozone and upwelling, coupled with both the large range of reanalysis upwelling trend estimates and the inability of WACCM-SD simulations to reproduce upwelling from their input reanalyses, severely limits the use of SD

simulations for accurately reproducing recent ozone variability. However, a free-running version of WACCM using only surface boundary conditions and a nudged QBO produces a positive decadal-scale lower stratospheric upwelling trend and a negative near-global lower stratospheric column ozone trend that is in closest agreement to the ozone observations.

## 1 Introduction

Due to the implementation of the Montreal Protocol on Substances that Deplete the Ozone Layer, the atmospheric concentration of ozone depleting substances (ODS) is declining, and ozone concentrations in the upper stratosphere and in the Antarctic ozone hole are exhibiting signs of recovery (WMO, 2022). The year of peak ODSs, as indicated by equivalent effective stratospheric chlorine (EESC), was in 1997, and since this time EESC has declined by about 9% to 17% for polar winter and midlatitude conditions (WMO, 2022). As such, studies considering ozone recovery have often analyzed the period

after 1997 during which ODSs were leveling off and beginning to decline.

Recently, an analysis by Ball et al. (2018, hereinafter B18) found that during this post-1997 period (1998-2016), satellite total column ozone (TCO) measurements indicated a flat trend in the near-global ($60°S – 60°N$) average. Using a separate set of vertically resolved satellite measurements, they further showed that the insignificant trend in TCO was the result of increases in the troposphere and upper stratosphere that were offset by decreases in the lower stratosphere. Ozone increases in the

troposphere and upper stratosphere were expected given the increases in tropospheric ozone pollution and the beginning of chemical ozone recovery in the upper stratosphere (respectively). However, the decrease in lower stratospheric ozone was a surprising result.

B18 investigated dynamical variability as a possible cause of the lower stratospheric ozone decline using specified dynamics (SD) simulations from the NCAR Community Earth System Model (CESM) Whole Atmosphere Community Climate Model

(WACCM) and the SOlar Climate Ozone Links (SOCOL) model. In these simulations, the model winds, temperature, and (for WACCM-SD) surface fields are nudged towards values provided by a reanalysis product such as MERRA/MERRA-2 (WACCM-SD) or ERA-Interim (SOCOL). Conceptually, the idea behind these types of simulations is that the reanalysis fields

used as input to SD simulations contain the dynamical variability that strongly determines ozone variability on interannual to decadal timescales. Somewhat surprisingly, B18 found that neither WACCM-SD nor SOCOL-SD reproduced the observed

lower stratospheric ozone decline. On the contrary, both models (and especially WACCM-SD) indicated significant positive trends in near-global (60°S – 60°N) stratospheric ozone column. Because of this, B18 suggested that the apparent decline in lower stratospheric ozone from 1998-2016 could be due to a non-dynamical process such as chemical ozone loss from halogenated very short-lived substances (VSLS), or to a deficiency in the representation of dynamical processes that control ozone in the SD model simulations.

Following B18, several studies have addressed the causes of short-term ozone variability and have concluded that dynamical factors rather than VSLS are the primary cause of recent lower stratospheric ozone changes. For example, Wargan et al. (2018) found that although lower stratospheric ozone decline in the MERRA-2 reanalysis and the MERRA-2 Global Modeling Initiative (M2-GMI) "replay" simulations (which are similar to nudging; see, e.g., Orbe et al., 2017) were consistent with the B18 results, the decline is consistent with enhanced two-way mixing between the tropics and midlatitudes. Chipperfield et al.

(2018) demonstrated that their chemical transport model (CTM) could accurately reproduce interannual ozone variability over the same period considered in B18, and using an additional year (2017) noted a strong rebound effect in near-global lower stratospheric ozone that largely cancelled out the apparent decline over the 1998 – 2016 period. However, more recently it was pointed out that the CTM results overpredict the 2017 rebound, and based on several additional years of data the robustness of the lower stratospheric ozone decline identified in B18 has been re-affirmed (Ball et al., 2019). Additional recent studies

have argued that the lower stratospheric trends are either not statistically significant (Godin-Beekmann et al., 2022), or are significant over the tropics in altitude coordinates but not in tropopause relative coordinates (and therefore dynamically driven, Thompson et al., 2021;Bognar et al., 2022). It is possible that the differences in significance among these studies are partially attributable to the differing statistical techniques (i.e., dynamical linear modeling vs. multiple linear regression, Bognar et al., 2022) or choice of regressors, but this has not been carefully assessed.

Independent of the robustness of the purported recent lower stratospheric ozone decline, these and other recent studies have raised important questions regarding the ability of SD simulations to faithfully reproduce the observed stratospheric dynamical

variability and its impacts on stratospheric composition (Ball et al., 2020;Chrysanthou et al., 2019;Dietmuller et al., 2021;Orbe et al., 2020). As an example, Davis et al. (2020) showed that the particular configuration of WACCM-SD used in B18 does not accurately reproduce the lower stratospheric upwelling trends present in the input (MERRA2) reanalysis over the past ~40 years (1980 – 2017). They investigated a series of alternative nudging schemes based on different combinations and aspects of horizontal winds and temperature and showed considerable sensitivity of the upwelling to the nudging configuration. A related study with the newest version of WACCM has demonstrated that the nudging timescale (i.e., the timescale over which the model field is relaxed towards the reanalysis fields) and reanalysis data frequency (e.g., 3-hourly versus 6-hourly) can also have a strong impact on the representation of stratospheric dynamical variability and tracer concentrations (Davis et al., 2022). Although no "silver bullet" SD configuration was identified, there were several robust findings coming out of these studies. First, spurious trends in upwelling may result from differences between the input reanalysis and model climatologies or from differing representations of gravity wave momentum forcing that drives a substantial fraction of upwelling. The former of these can be somewhat ameliorated by nudging the anomalies of the wind/temperature variables rather than the actual fields themselves. Of particular concern is nudging the actual (zonal mean) temperatures, as it can paradoxically drive nonsensical temperature trends and spurious wave forcings (Davis et al., 2020). Another robust finding (at least for WACCM) is that the dynamical representation is not particularly sensitive to nudging timescale, but is somewhat sensitive to the frequency of the input meteorological fields (Davis et al., 2022).

Although many recent studies have demonstrated that dynamical processes are key in driving decadal scale ozone variations, the specific reasons for the discrepancy between the SD-simulated ozone increases and satellite-observed lower stratospheric decline over 1998 – 2016 in B18 remain unexplained. The goal of this paper is to identify the cause of this discrepancy in order to aid in the interpretation of the recent satellite record and potentially provide guidance for modeling studies attempting to simulate interannual to decadal scale ozone variability and the dynamical processes that drive them. We revisit the WACCM-SD simulation used in B18 to ask whether or not it contains the same underlying dynamical variability as the input reanalysis it uses. We also consider additional SD and free-running WACCM simulations in order to diagnose the cause of the mismatch between simulated and observed short-term ozone trends.

## 2 Methods and Data

In this paper, we use simulations of the NCAR Whole Atmosphere Community Climate Model (WACCM, Marsh et al., 2013) version 1.2.2, which is part of the NCAR Community Earth System Model (Hurrell et al., 2013). Specifically, we use the suite of free-running and specified dynamics (SD) atmosphere-only WACCM simulations described in Davis et al. (2020) and summarized briefly below (see also Table 1). All simulations are run with prescribed historical sea surface temperatures and sea ice concentrations (Hurrell et al., 2008) at 1.9° x 2.5° horizontal resolution and either 66 levels (corresponding to the free-running version of the model) or 88 levels (72 MERRA-2 levels plus 16 additional levels above the MERRA-2 lid at 0.01 hPa) spanning 0 – 140 km. The WACCM L66 grid is virtually identical to MERRA2 between 72 hPa and 266 hPa (inclusive), and has a vertical resolution of 1 km in this region. The WACCM-SD simulations nudge surface and atmospheric variables in grid-point space with a 50 hour timescale (1% per 30 minute time step) towards their corresponding fields using 3-hourly output from the MERRA-2 reanalysis (collection M2I3NVASM, GMAO, 2015;Gelaro et al., 2017). For 3D fields, nudging is applied uniformly at all levels below 0.8 hPa, and is linearly reduced from 1% per time step to 0% between 0.8 hPa (~50 km) and 0.2 hPa (~60 km).

The WACCM-SD simulation with the default configuration ("UVT L88", see Table 1) using MERRA-2 vertical levels is the same simulation used in B18 and Davis et al. (2020), and whose configuration described in Lamarque et al. (2012). In addition to simulations with the default SD configuration, we also use 66-level SD simulations as in Davis et al. (2020) in which MERRA-2 has been interpolated to WACCM's native vertical grid. The motivation for these simulations is to reduce spurious latent heating and gravity wave momentum forcing due to parameterizations for convection and gravity wave generation, respectively. These parameterizations respond dramatically differently when used with the L88 vertical grid, as the L88 grid contains approximately triple the number of vertical levels in the boundary layer and lower troposphere.

Several additional SD simulations with alternative configurations are included here, including several nudging only winds (denoted "UV"), as well as several that nudge "climatological anomalies" and "zonal anomalies" of the reanalysis variables

(see Davis et al., 2020 for details of these configurations). Additionally, we consider one SD simulation on native WACCM

levels (L66) using the winds and temperature from the ERA-Interim reanalysis (UVT L66 ERAI) to test whether the results

are sensitive to the choice of reanalysis.

Additionally, we also use two free running atmosphere-only WACCM simulations over the same period as MERRA-2 (i.e.,

starting in 1980). One simulation is the default WACCM configuration that nudges the equatorial (5°S – 5°N) stratospheric

winds (100 hPa – 10 hPa) to MERRA-2 in order to produce a quasi-biennial oscillation ("AMIPQBO"), and the other

simulation uses no nudging and has persistent easterlies in the equatorial stratosphere with no internally generated QBO

(AMIPnoQBO).

TABLE 1. Summary of WACCM simulations

| Short name | Description | Nudged atmospheric variables | Vertical levels |
|---|---|---|---|
| AMIPnoQBO | Free running (no QBO) | None | 66 |
| AMIPQBO | Free running (nudged QBO) | u (tropics only) | 66 |
| UVT L88 | SD default configuration | u, v, T | 88 (MERRA2 levels) |
| UV L88 | SD default configuration (no T nudging) | u, v | 88 (MERRA2 levels) |
| UVT | SD on free-running model levels | u, v, T | 66 |
| UV | SD on free-running model levels (no T nudging) | u, v | 66 |
| $U_{ca}V_{ca}T_{ca}$ | SD climatological anomaly | u, v, T climatological anomalies | 66 |
| $U_{ca}V_{ca}$ | SD climatological anomaly (no T nudging) | u, v climatological anomalies | 66 |
| $U_{za}V_{za}T_{za}$ | SD zonal anomaly | u, v, T zonal anomalies | 66 |
| $U_{za}V_{za}$ | SD zonal anomaly (no T nudging) | u, v zonal anomalies | 66 |
| UVT ERAI | SD on free-running model levels w/ ERA-I | u, v, T | 66 |

All WACCM simulations include online chemistry, and we have additionally included online diagnostic calculations of the

Transformed Eulerian Mean (TEM) velocities and tracer budget terms for ozone (eq. 9.4.13, Andrews et al., 1987). The tracer

budget equation for species $\overline{\chi}$ is

$$\overline{\chi}_t = -\overline{v}^* \overline{\chi}_y - \overline{w}^* \overline{\chi}_z + \rho_0^{-1} \nabla \cdot \rho_0 \boldsymbol{M} + P - L \qquad\qquad (1)$$

where subscripts denote meridional (y), vertical (z) and time (t) derivatives; $\overline{v}^*$ and $\overline{w}^*$ are the TEM residual mean velocities;

$\boldsymbol{M}$ is the eddy flux vector; and P and L are the chemical production and loss terms.

  Monthly means of the ozone advection, mixing, and production/loss tendencies are stored for analyzing contributions to ozone variability from various processes. For analyzing tropical upwelling, we also compute the tropical upwelling mass flux, M* (e.g., as in Rosenlof, 1995), by differencing the extrema in each hemisphere of the TEM streamfunction based on the meridional velocity $(\overline{v}^*)$. We note that M* can be quantified as a function of pressure.

We also analyze tropical upwelling from the MERRA2 reanalysis (Gelaro et al., 2017) that is used as input to the WACCM-SD simulations (collection M2I3NVASM, GMAO, 2015). Additionally, we analyze output from the ERA-Interim (ERA-I) reanalysis (Dee et al., 2011), which is used as the input for UVT ERAI simulation. As a check against the most recent version of the reanalysis, we also include the ERA-5 reanalysis (Hersbach et al., 2020). The reanalysis TEM velocities used here come from the dataset described in Martineau et al. (2018). For computing tropical upwelling mass flux from reanalysis, we use the

TEM streamfunction based on the meridional velocity.

  To quantify changes in ozone and other variables, we use both ordinary least squares fits as well as the same dynamical linear modeling (DLM) technique (Laine et al., 2014) and regressors used in B18 and implemented in the `dlmmc` software package described by Alsing (2019). Briefly, the DLM is similar to commonly-used multiple linear regression (MLR) in that it accounts for variability in the ozone fields using linear regressors, but unlike MLR it can include autoregressive terms and slowly-

varying seasonal cycle and trend terms. The optimal DLM model parameters are inferred using Hamiltonian Monte Carlo sampling and a Kalman filter. In this paper, we use the `dlm_vanilla_ar1` model provided by `dlmmc`, which includes a first-order autoregressive term and fixed (non-time varying) regressor terms. Of most relevance here is the posterior distribution of the trend term and the sampled distribution of the difference between the start and end of the period considered here (i.e., 1998 – 2016). As in B18, for the DLM analysis we use the following regressors: the Singapore winds at 30 and 50

hPa, the F30 solar flux, the Oceanic Niño Index, stratospheric aerosol optical depth, and the Arctic and Antarctic Oscillation indices. Also as in B18, we use the percentage of posteriors that are of a given sign (e.g., 95%) as a measure of significance of

the trend. The DLM is applied to ozone fields from both the WACCM simulations and the merged zonal mean satellite ozone product from the Stratospheric Water and Ozone Satellite Homogenized (SWOOSH) version 2.6 data (Davis et al., 2016). Finally, for analyses using simple linear trends, we provide trend uncertainties at the 95% confidence interval and significance

at the 95% level with a two-sided student's t-test, both accounting for autocorrelation in the residuals. Analogous considerations are taken into account regarding autocorrelation when determining the statistical significance of correlation coefficients.

## 3 Analysis

### 3.1 Dynamical variability over 1998 - 2016

We first consider dynamical and ozone variability in the WACCM simulations over the 1998 – 2016 period, the same period considered in B18. Figure 1 shows the tropical upwelling mass flux (M*) anomaly timeseries at 85 hPa for the suite of WACCM simulations. The 85 hPa level is the first level above the climatological tropical tropopause in WACCM, and, as will be shown later in the paper, is the level most closely associated with ozone variability. As can be seen in Fig. 1, there are large differences in both the interannual variability and trends between the two reanalyses, the WACCM-SD simulations, and the

two free-running versions of WACCM. The reanalysis upwelling timeseries are generally well correlated with one another, with some notable differences around 2010. While ERA-I contains a positive trend ($3.7 \pm 2.8$ % decade$^{-1}$), there is no significant trend in MERRA2 upwelling during the time period considered by B18. In contrast, the WACCM-SD simulation used in B18 (UVT L88) contains a negative trend in lower stratospheric upwelling ($-3.0 \pm 2.3$ % decade$^{-1}$). Given that the climatological vertical gradient of ozone in the tropical lower stratosphere is negative, the slowdown in upwelling in the UVT

L88 simulation used by B18 is consistent with an increase in the modeled lower stratospheric ozone concentrations observed in B18. Other SD configurations either have statistically insignificant trends or negative trends (i.e., UVT and UV).

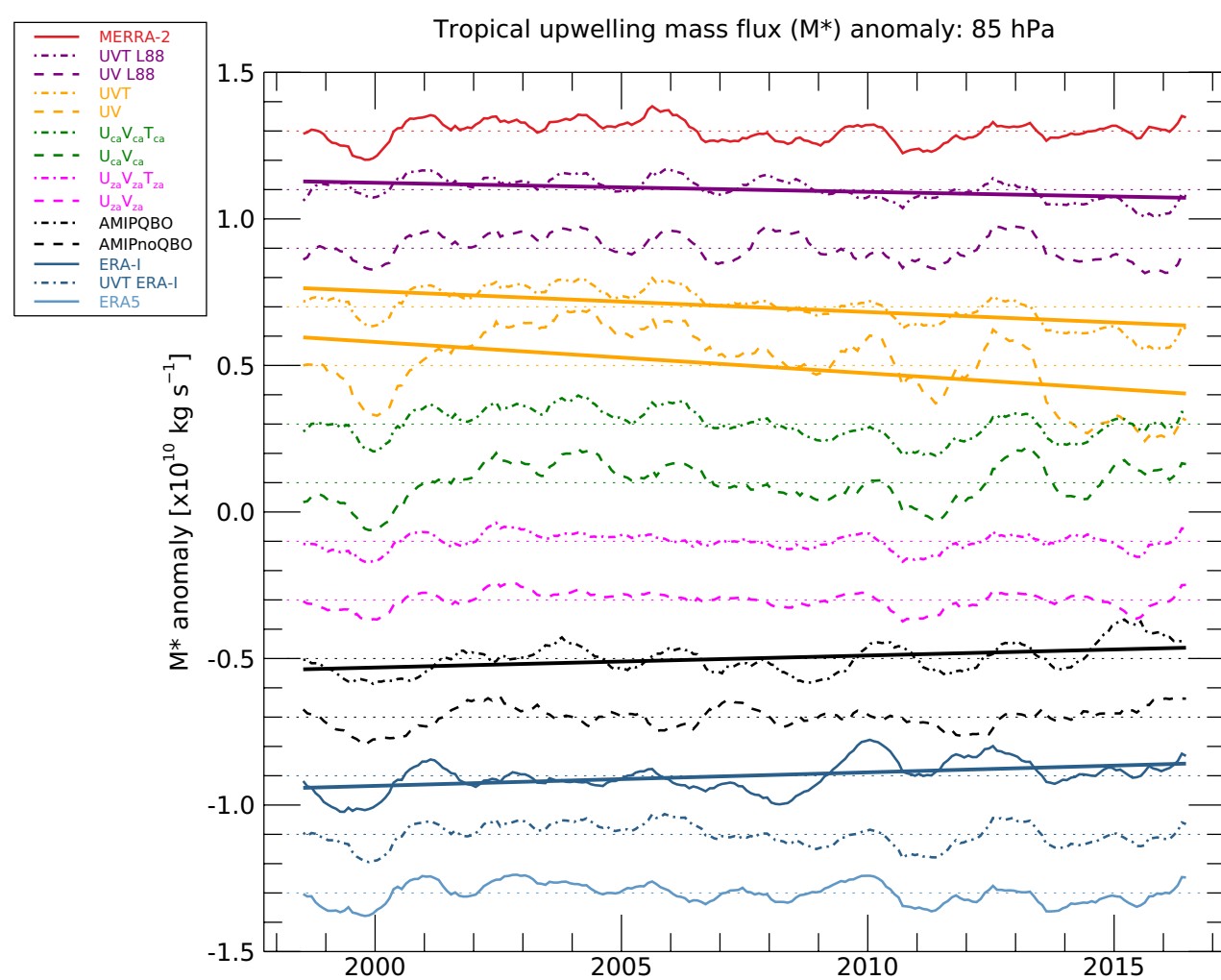

**Figure 1: Timeseries of 13-month smoothed tropical upwelling mass flux (M\*) anomalies at 85 hPa over 1998-2016 in WACCM SD and free-running simulations (see Table 1 for simulation names), and for the MERRA-2 and ERA-I reanalyses. Data are offset in increments of 0.2 x 10$^{10}$ kg s$^{-1}$ for clarity. Dotted lines show the zero value for each timeseries, and trend lines are indicated for simulations with statistically significant trends.**

As expected, the free-running simulations do not closely follow the reanalyses timeseries due to unconstrained atmospheric variability. The free-running simulation without a nudged QBO (AMIPnoQBO) contains significantly less upwelling variability than the one with the QBO (AMIPQBO), highlighting the importance of the QBO in driving interannual variability in upwelling. Additionally, the AMIPQBO simulation has a positive upwelling trend over the B18 period ($3.7 \pm 2.8$ % decade$^{-}$

[1]), whereas the AMIPnoQBO trend is not statistically significant. This behavior suggests that over short time periods upwelling trend estimates are affected by QBO-related variability (e.g., the QBO-related increase in 2015/2016), and are perhaps sensitive to the QBO phase at the starting and ending points of the timeseries. It is worth noting that all of the SD configurations that nudge towards the historical QBO winds (i.e., all except for the $U_{za}V_{za}T_{za}$ and $U_{za}V_{za}$ simulations) capture some QBO-like variability in their upwelling (see Fig. 1).

To quantify the degree to which upwelling is reproduced in the SD simulations over the 1998 – 2016 time period, Figure 2 shows the trends and the correlation in tropical upwelling between each WACCM SD simulation and the reanalysis it uses as input. For reference, the inter-reanalysis correlations are shown, as well as the correlations between free-running simulations and MERRA-2. As expected, neither of the free-running simulations correlate strongly with the reanalyses, but the AMIPQBO simulation is better correlated than AMIPnoQBO above 30 hPa.

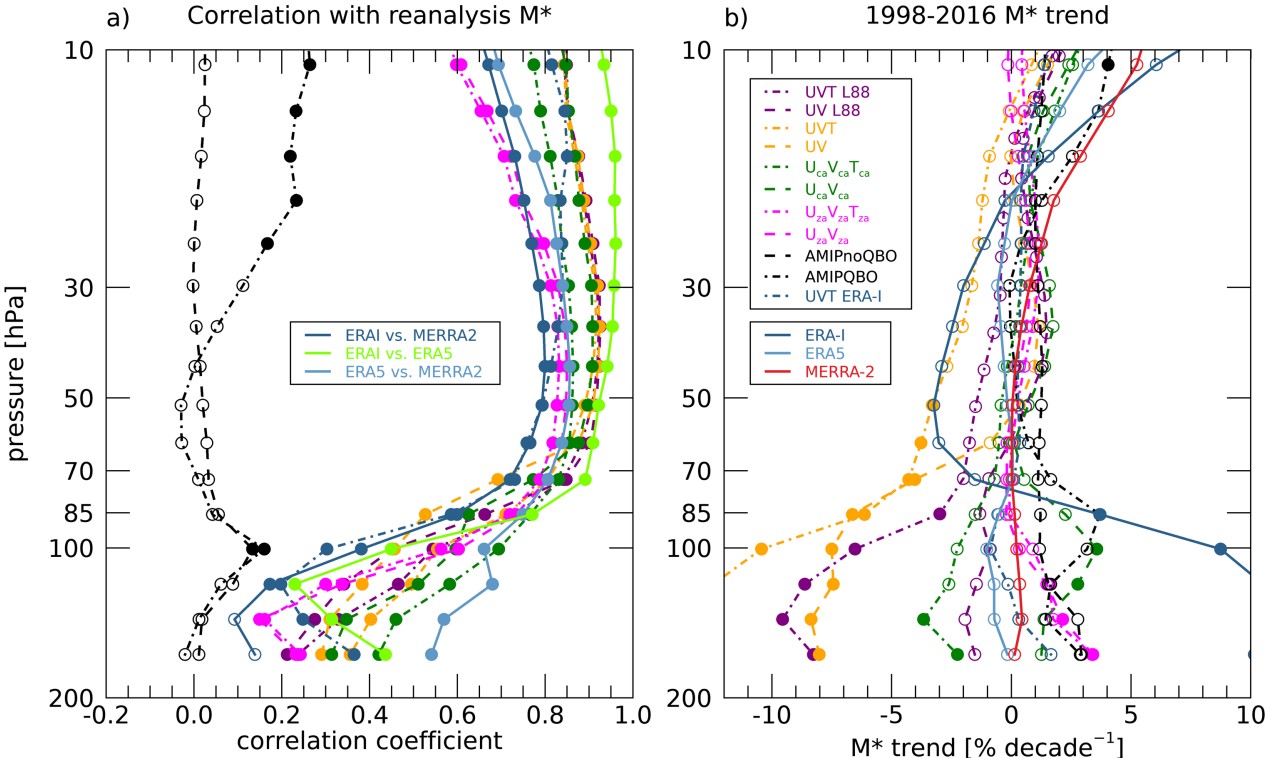

**Figure 2: (a) Correlation of monthly tropical upwelling mass flux (M\*) anomalies (1998 – 2016) as a function of pressure between each of the WACCM simulations and the MERRA-2 reanalysis (except for the UVT ERA-I simulation, for which correlations with**

**ERA-I are shown). (b) Tropical upwelling mass flux trend as a function of pressure for the WACCM simulations and reanalyses. In each panel, statistically significant correlations and trends are marked with filled circles.**

In general, the SD simulations show the best correlation with their input reanalysis around 50 hPa and above, with a reduction in the correlation as one goes down in altitude from this level. The inter-reanalysis correlations exhibit a similar behavior with

height, with the best correlated pair in the lower stratosphere being the two most recent reanalyses (ERA5 and MERRA2). The relatively poor agreement among the different reanalyses in the lower stratosphere is consistent with previous work showing large disagreements in this region (Randel and Jensen, 2013;Wright et al., 2020). Overall, while the correlation between SD simulations and their input reanalysis upwelling is quite good in the mid and upper stratosphere, the weaker correlations in the lower stratosphere support the notion that these simulations may have difficulty in reproducing the variability of trace gases

such as ozone that are dynamically controlled. Also, it is worth noting that the climatological anomaly simulations (green in Fig. 2a) have a relatively high correlation in the lower stratosphere, suggesting that nudging the climatological fields below ~85 hPa adds little to no value and in fact may be detrimental to reproducing upwelling.

The three reanalyses mostly show insignificant trends in the stratosphere over 1998 – 2016, but exhibit some notably different behavior. ERA-I upwelling trends show greater variation with height than MERRA-2 and ERA5, as well as large positive

trends at and below 85 hPa, indicating a strengthening shallow branch and weakening deep branch of the Brewer-Dobson circulation. In contrast, ERA5 trends are all insignificant, and MERRA-2 trends are mostly insignificant and lie between 0 and 5% decade$^{-1}$ for all but the uppermost point.

For the SD simulations above 70 hPa, the upwelling trends are mostly insignificant and cluster between approximately ± 3% decade$^{-1}$. Below this level, there is significantly more spread in the trends. Notably, the UVT L88 simulation used in B18

shows significant negative trends below 70 hPa, which are consistent with positive trends in tropical ozone, at least at these levels. Overall, the SD simulations on the (66) native WACCM model levels (UVT/UV) show the strongest negative upwelling trends among all of the SD simulations. Other than these three simulations that have negative trends in the lower stratosphere,

the other simulations show a variety of behavior, with mostly insignificant trends within a range of approximately $\pm 4\%$ decade[1].

## 3.2 Ozone trends over 1998 – 2016

We have so far established that the SD simulation used by B18 exhibits a significant slowdown in lower stratospheric tropical upwelling over 1998 – 2016, in contrast to the (insignificant) trend present in the MERRA-2 reanalysis it uses as input, and that the suite of SD and free-running simulations produce a variety of different behavior in the lower stratosphere. We next turn our attention to the ozone trends present in these simulations before investigating how the ozone and upwelling trends might be related. Figure 3 shows the ozone trends as function of latitude and height for the full suite of simulations and SWOOSH. We do not include trends of the reanalysis ozone fields themselves due to known inhomogeneities associated with the assimilation of vertically resolved ozone observations (Wargan et al., 2018;Davis et al., 2017).

As found in B18, the UVT L88 simulation shows a significant increase in ozone in the lower stratosphere over a broad range of latitudes (60°S – 60°N), whereas over this same region the observations (in this case, SWOOSH) show a significant decrease. Lower stratospheric ozone increases of varying degrees are found across the other SD simulations that nudge the full wind/temperature fields (i.e., UV* and UVT* simulations), whereas decreases are present in the free-running simulations, zonal anomaly nudging simulations, and $U_{ca}V_{ca}$ simulations. We note that simply nudging the temperature anomalies produces a change from negative ozone trends (i.e., in $U_{ca}V_{ca}$) to positive ($U_{ca}V_{ca}T_{ca}$). This trend reversal does not happen in the zonal anomaly cases, where the zonal mean temperature is not explicitly being nudged.

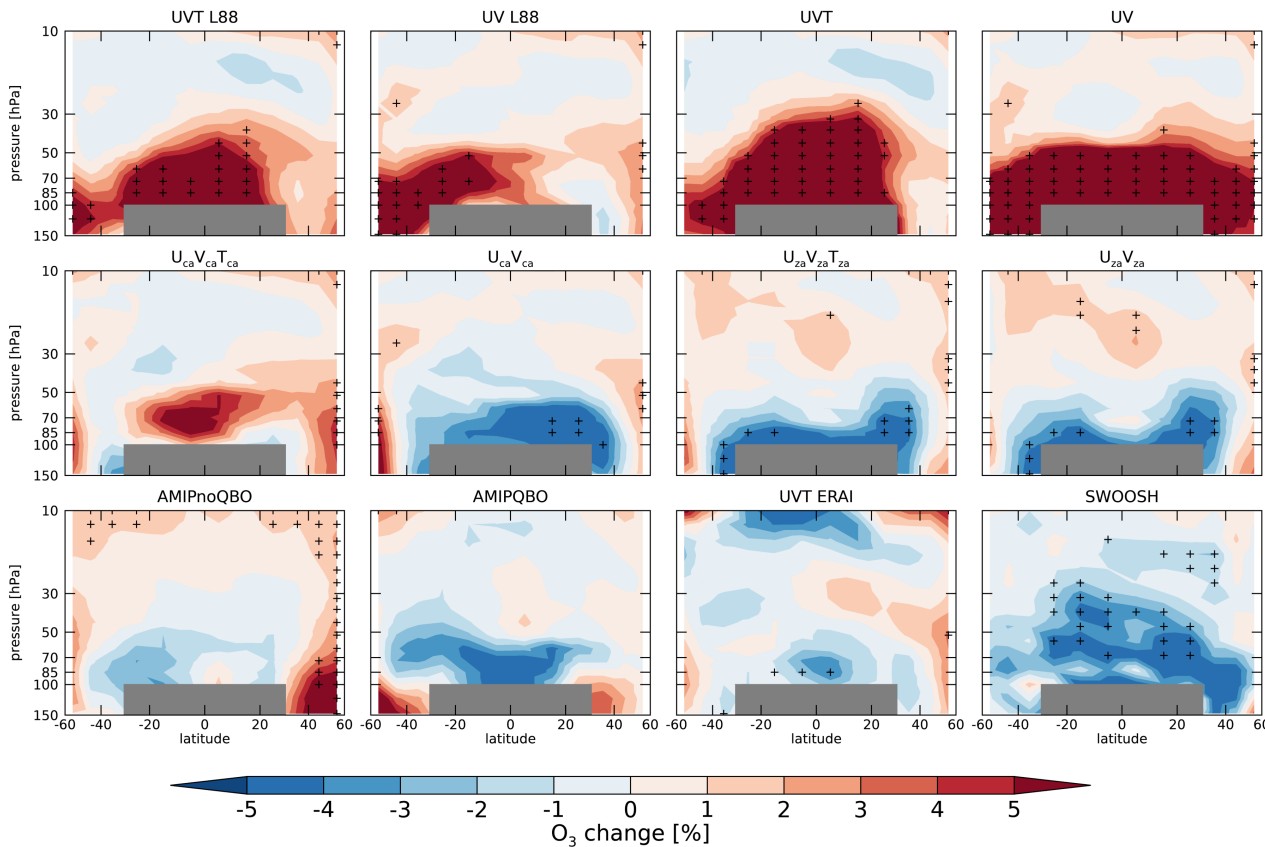

**Figure 3: Percent change in ozone in WACCM simulations and SWOOSH data over the period 1998 – 2016, based on the dynamical linear model used in B18 applied to 10° binned data, and comparable to Fig. 5 of B18. Statistically significant changes are denoted by plus symbols. Gray areas denote the region excluded from the near-global lower stratospheric ozone columns presented elsewhere in the paper.**

So far, the results have confirmed the findings of B18 that the standard WACCM SD configuration is unable to reproduce the observed negative lower stratospheric ozone trends or their pattern (e.g., the enhanced region extending to NH midlatitudes). However, the free-running simulation and several of the alternative configurations proposed by Davis et al. [2020] show hints of a lower stratospheric ozone decline, as does the simulation using the ERA-Interim reanalysis winds/temperatures. In the next sections, we more closely consider the dynamical variability and how it may relate to spread in ozone trend behavior among the various simulations.

### 3.3 Correlation between ozone and upwelling: month-to-month variability

We first consider the correlation between lower stratospheric partial column ozone and tropical upwelling at various levels in

the WACCM simulations. Here, we are attempting to identify the level at which tropical upwelling mass flux is most highly

correlated with near-global lower stratospheric column ozone (hereinafter LSCO), defined (as in B18) as the 60°S – 60°N

average partial column over 100 hPa to 30 hPa for latitudes equatorward of 30°, and 147 hPa to 30 hPa for latitudes between

30° and 60° in each hemisphere. To answer this, Figure 4 shows correlations between LSCO and tropical upwelling mass flux

at different levels. As might be expected if anomalous increases in upwelling lead to increased advection of ozone-poor air

into the stratosphere, there is a negative correlation between ozone and upwelling in the lower stratosphere around 100 hPa.

In other words, months with above average tropical upwelling near 100 hPa tend to be months with less ozone integrated over

a broad geographic and vertical region of the stratosphere. For all simulations, the most negative correlations between ozone

and upwelling occur between 120 and 85 hPa. All simulations also show correlations closer to zero (or even slightly positive)

with increasing height due to the decreased chemical lifetime at higher altitudes. It is not obvious that LSCO averaged over

60°S – 60°N should be negatively correlated with tropical upwelling in the lower stratosphere. While it is to be expected that

equatorial column ozone should correlate negatively with lower stratospheric upwelling, the latitude band considered here and

in B18 (60°S – 60°N) includes latitudes with significant ozone downwelling and mixing, and these processes could potentially

lead to positive correlations between stratospheric upwelling and LSCO at midlatitudes. Considering just tropical latitudes

30°S – 30°N (Fig. S1), the anti-correlation between upwelling and LSCO is stronger and the sensitivity of LSCO to upwelling

around 85 hPa is greater than over 60°S – 60°N.

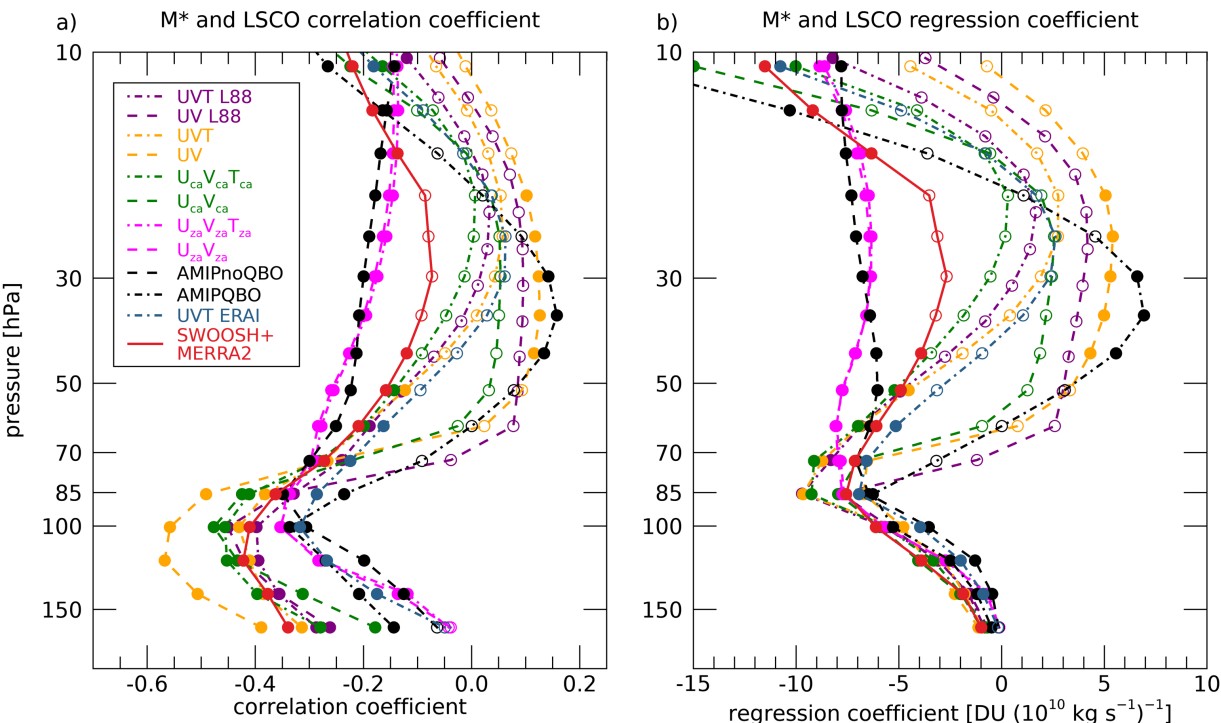

**Figure 4: (a)** The correlation between the monthly (deseasonalized) anomalies of near-global lower stratospheric ozone column (LSCO) and tropical upwelling mass flux (M*) at different levels for each WACCM simulation (1980 – 2016). LSCO is defined (as in B18) as the 60°S – 60°N average partial column over 100 hPa to 30 hPa for latitudes equatorward of 30°, and 147 hPa to 30 hPa for latitudes between 30° and 60° in each hemisphere. Also shown are the values for SWOOSH LSCO and tropical upwelling mass flux anomalies from MERRA2 (red line, 1984 – 2016). Significant (insignificant) correlations are shown as filled (open) circles. **(b)** The regression coefficients from a linear fit of tropical upwelling mass flux anomalies at each vertical level to LSCO anomalies.

It is also worth noting that while most of the simulations look quite similar in their behavior, there are a few notable outliers. The AMIPnoQBO and zonal anomaly simulations both show consistently negative correlations between upwelling and LSCO above 60 hPa, and do not exhibit a broad positive peak around 20 – 40 hPa that is present in most other simulations. This differing behavior is explained by the lack of a nudged QBO in these simulations, presumably due to their lack of anti-correlation between upwelling in the lower stratosphere (e.g., at 85 hPa) and mid-stratosphere (e.g., at 30 hPa) that is related to the QBO secondary circulation.

Figure 4 also includes an "observational" estimate based on MERRA2 upwelling and SWOOSH LSCO. For both the correlation and regression coefficients, in the lower to middle stratosphere the SWOOSH/MERRA2 values look similar to the SD simulations and the free-running simulation with QBO. Of course, the MERRA2 meteorological fields are used in most of the SD simulations, so the observational estimate should not be interpreted as an independent validation for those simulations.

That said, the observational estimate as well as all of the model simulations agree that variations in upwelling at 85 hPa have the largest impact on LSCO (Fig. 4b), with little variability across configurations within the TTL.

Given that variations in 85 hPa tropical upwelling mass flux are most closely associated with changes in near-global lower stratospheric ozone column, we next seek to investigate where (vertically and horizontally) the ozone is most closely correlated with upwelling variability at this level. Figure 5 shows the regression coefficient of a linear fit of tropical upwelling mass flux

anomalies at 85 hPa to ozone partial column anomalies for each of the simulations. As can be seen from this figure, the tropical lower stratosphere is the region where ozone is most sensitive to upwelling. For all of the simulations, the majority of the domain over which the near-global lower stratospheric ozone column is computed (i.e., 60°S – 60°N, 147/100 – 30 hPa) contains negative regression coefficients, in agreement with the negative correlation coefficients found for 85 hPa upwelling in Fig. 4.

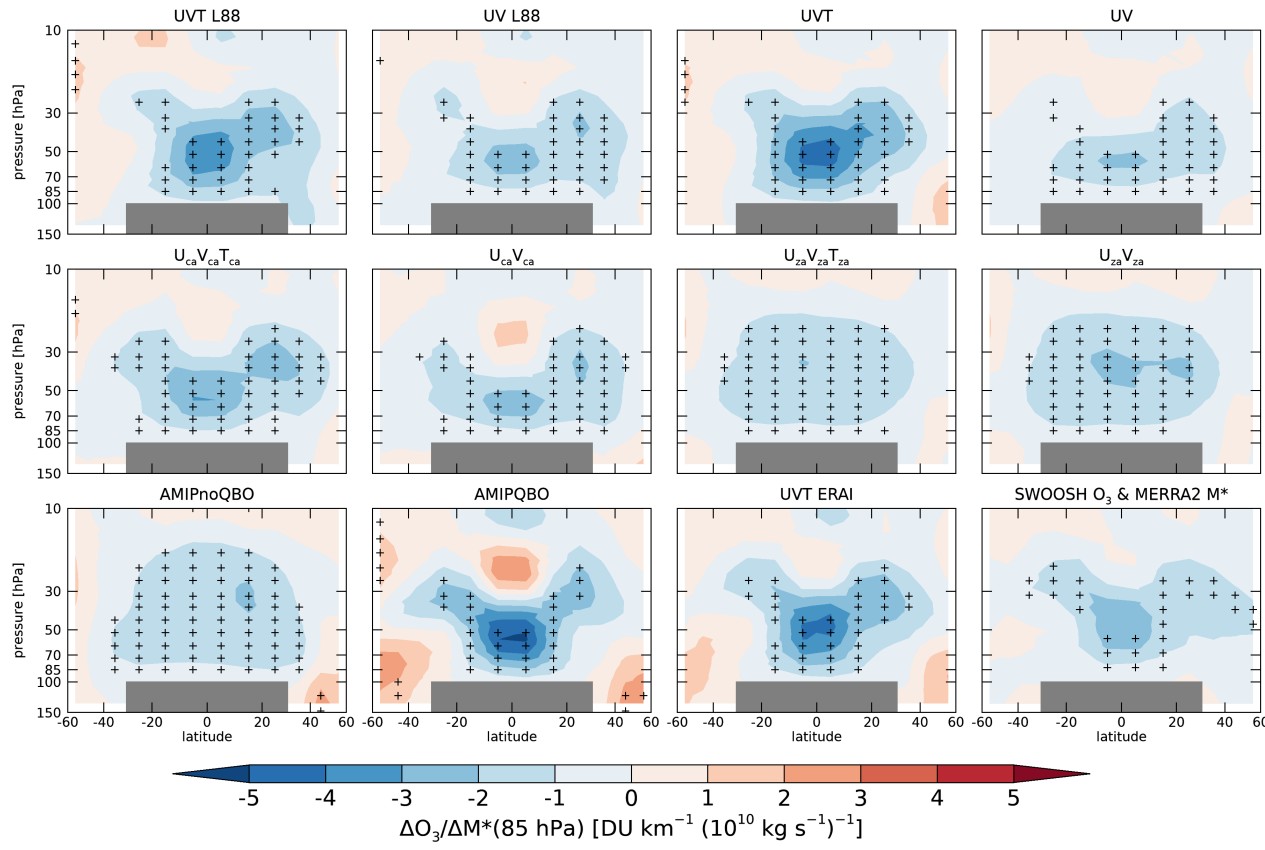

**Figure 5: Regression coefficients from a linear fit of monthly stratospheric partial column ozone anomalies to tropical upwelling mass flux anomalies at 85 hPa for the model simulations (1980 – 2016) and observations (SWOOSH ozone and MERRA2 upwelling, 1984 – 2016). The horizontal scale is area weighted, so the area and magnitude of the contours below 30 hPa is roughly representative of the LSCO anomalies presented elsewhere. Statistically significant changes are denoted by plus symbols.**

There are several differences among the regression patterns for the different simulations that are worth noting. First, simulations lacking a nudged QBO (i.e., the zonal anomaly simulations and AMIPnoQBO simulation) contain a broad region of relatively weak negative coefficients that spans $\pm \sim 40°$ around the equator and extends upwards to nearly 10 hPa. In contrast, simulations with a nudged QBO tend to show a "bird-shaped" pattern consisting of more negative coefficients (relative to non-QBO simulations) in a narrower region around the equator below ~30 hPa with "wings" extending poleward and upward, as well as a region of positive coefficients located directly above the negative region. Considering the behavior of the non-

nudged QBO simulations in Figs. 4 and 5, it is apparent that these simulations behave significantly differently than the rest of the simulations that nudge towards a QBO. The reasons for these differences are related to the QBO secondary circulation, and will be explored further in section 3.5.

Another notable difference among the simulations is that not nudging temperature leads to a weaker sensitivity of ozone to upwelling in the tropical lower stratosphere, as evidenced by the difference between the two pairs of UV/UVT simulations in the top row of Fig. 5. As free-running WACCM is colder than MERRA2 in the tropical lower stratosphere and has a ~1 km higher tropopause (Davis et al., 2020), it makes sense that including nudging to MERRA2 temperatures acts like diabatic heating and enhances upwelling, compared to configurations that just nudge horizontal winds. However, it is not immediately

obvious why ozone should vary more (per unit upwelling change) in temperature-nudged simulations than in non-temperature-nudged simulations. A possible explanation is that while upwelling is mostly driven by resolved wave drag in the model (and is thus strongly affected by wind nudging), wave drag also induces horizontal mixing and thus complicates the ozone response. In contrast, temperature nudging may more directly invigorate upwelling, leading to a stronger ozone response per unit upwelling change as seen in Fig. 5.

**3.4 Correlation between ozone and upwelling: trends**

Given the correlation between upwelling and LSCO, we next turn our attention to how the trends in these quantities are related. Figure 6 demonstrates the high degree of correlation between tropical upwelling trends at 85 hPa and the 60°S – 60°N LSCO trends amongst the different model simulations (r = -0.93), and also shows the reanalysis upwelling trends as well as observed LSCO changes from B18 and SWOOSH. Over the tropical latitudes (Fig. S5), the correlations are similar (r = -0.87), with

weaker correlations in the NH (Fig. S6, r = -0.60) and SH (Fig. S7, r = -0.82).

A linear fit to the model simulation data in Fig. 6 gives a y-intercept of $0.3 \pm 0.1$ DU decade$^{-1}$, suggesting that a simulation with no upwelling trend would produce a small increase in LSCO. In the case of the simulation used by B18 (i.e., UVT L88), the negative trend in upwelling in that simulation appears to explain roughly half of the discrepancy between the modeled and observed LSCO changes. In other words, based on the relationship in Fig. 6, if the UVT L88 (which contains an LSCO trend

of ~1.2 DU decade[-1]) simulation properly reproduced the MERRA2 upwelling trend, the LSCO ozone trend would be near zero. In contrast, the SWOOSH and B18 trends are ~ -1 DU decade[-1].

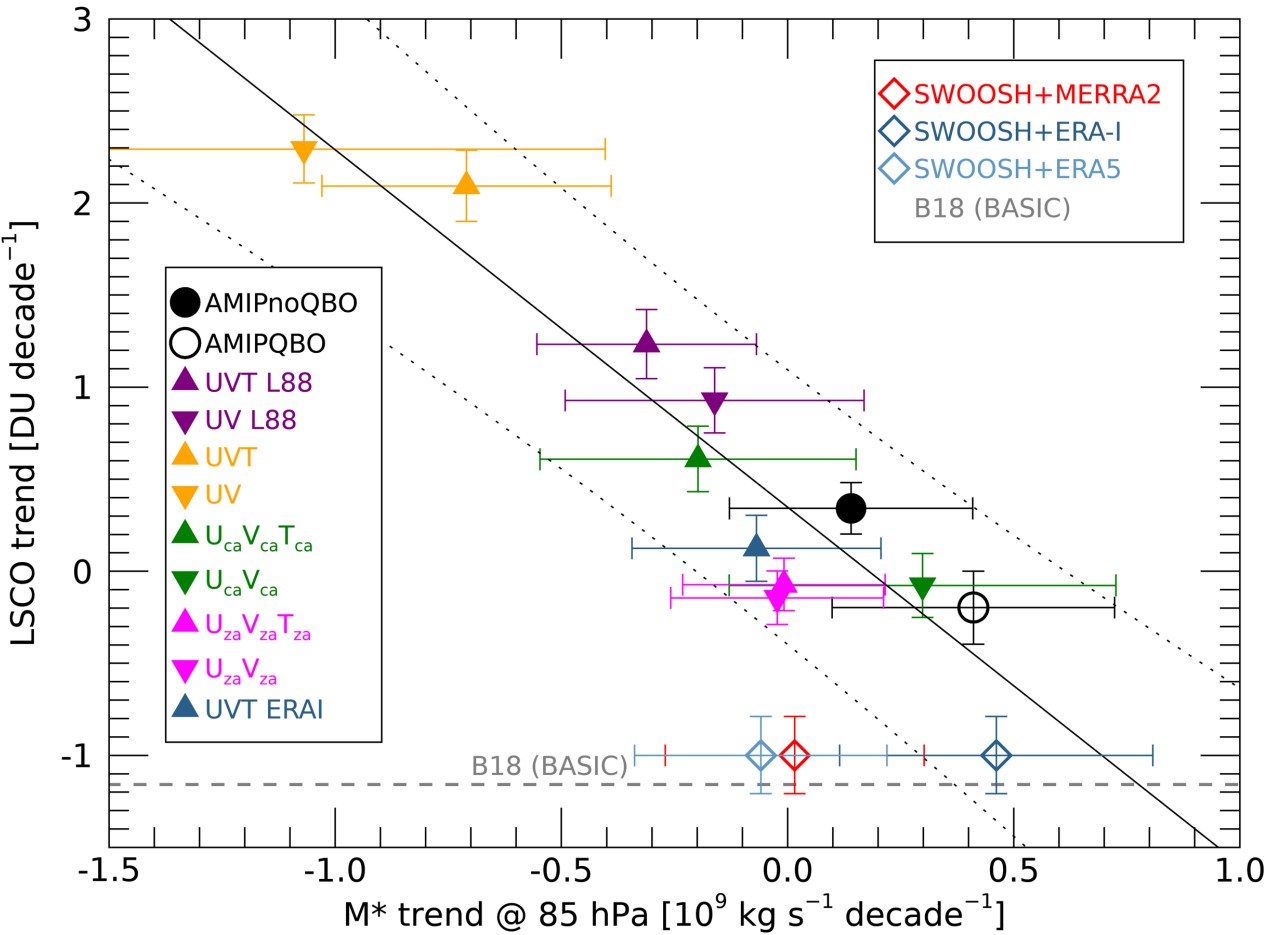

**Figure 6: Near-global (60°S – 60°N) lower stratospheric column ozone (LSCO) trends over 1998-2016 plotted against tropical upwelling mass flux trends at 85 hPa for a suite of WACCM simulations (triangles, specified dynamics; circles, free-running). Also shown (diamonds) are estimates from the SWOOSH data and various reanalyses that are used in the SD simulations. The horizontal dashed line shows the equivalent estimate of ozone change from B18 over this time period, based on the BASIC-SG data set. The solid line is the linear fit to the WACCM simulations, and the dotted lines denote the 95% prediction interval.**

Despite upwelling trends appearing to explain most of the spread in LSCO trends amongst the simulations, several issues remain. First, the observational LSCO estimates based on SWOOSH and MERRA2/ERA5 fall well below the spread of the

model simulations as quantified by the 95% prediction interval (dotted lines in Fig. 6). In other words, even for SD simulations that accurately reproduce the upwelling trends in MERRA2/ERA5 (e.g., $U_{za}V_{za}T_{za}$), their simulated ozone trends are significantly more positive than the observed ozone trends from SWOOSH and BASIC-SG.

One caveat is that the ERA-I upwelling trend does lie within the prediction interval in Fig. 6, suggesting that if the UVT ERAI

simulation accurately reproduced the ERA-I upwelling trend at 85 hPa, it would produce an LSCO trend consistent with observations. That said, previous work has documented a ~40% positive bias in mean state TTL upwelling in ERA-I (Ploeger et al., 2021;Ploeger et al., 2012) and a disagreement between its long-term upwelling trend at 70 hPa and other data sources when using the standard TEM formulation based on winds and temperature  (Diallo et al., 2021;Ploeger et al., 2021;Ploeger et al., 2012;Seviour et al., 2012;Abalos et al., 2015).

Also, while the strong correlation shown in Fig. 6 is suggestive of a causal relationship where increased tropical lower stratospheric upwelling causes ozone decreases (and vice versa), it is not immediately clear whether the response is due to advection or mixing. At least in the tropics, there are large offsetting contributions to the ozone budget from photochemical production/horizontal mixing and vertical advection (Abalos et al., 2013;Abalos et al., 2012). In the next section, we consider the changes in the ozone budget terms in an attempt to identify the cause of the modeled ozone changes.

**3.5 Ozone budget analysis**

Using the TEM tracer budget terms, we assess here how the various chemical and physical processes affecting ozone concentrations relate to tropical upwelling in the lower stratosphere. Figure 7 illustrates that, as expected, the ozone tendency due to advection (i.e., first and second terms in eq.1) is negative for anomalous positive tropical upwelling at 85 hPa. The equivalent figure for the mixing tendency (Figure 8) shows generally the opposite behavior, with anomalous positive upwelling

at 85 hPa being associated with decreases in mixing along the flanks of the "tropical pipe". Similarly, the net chemical tendency (Figure 9) tends to show opposite behavior to the advective tendencies, with a broad region where increased net production is correlated with anomalous upwelling. Together, these figures (along with Fig. 5) illustrate that increased upwelling is associated with negative ozone advective tendencies that are coherent over a broad geographical and vertical region throughout

the stratosphere, and that these negative advective tendencies dominate over the (often opposite-signed) mixing and chemical

tendencies.

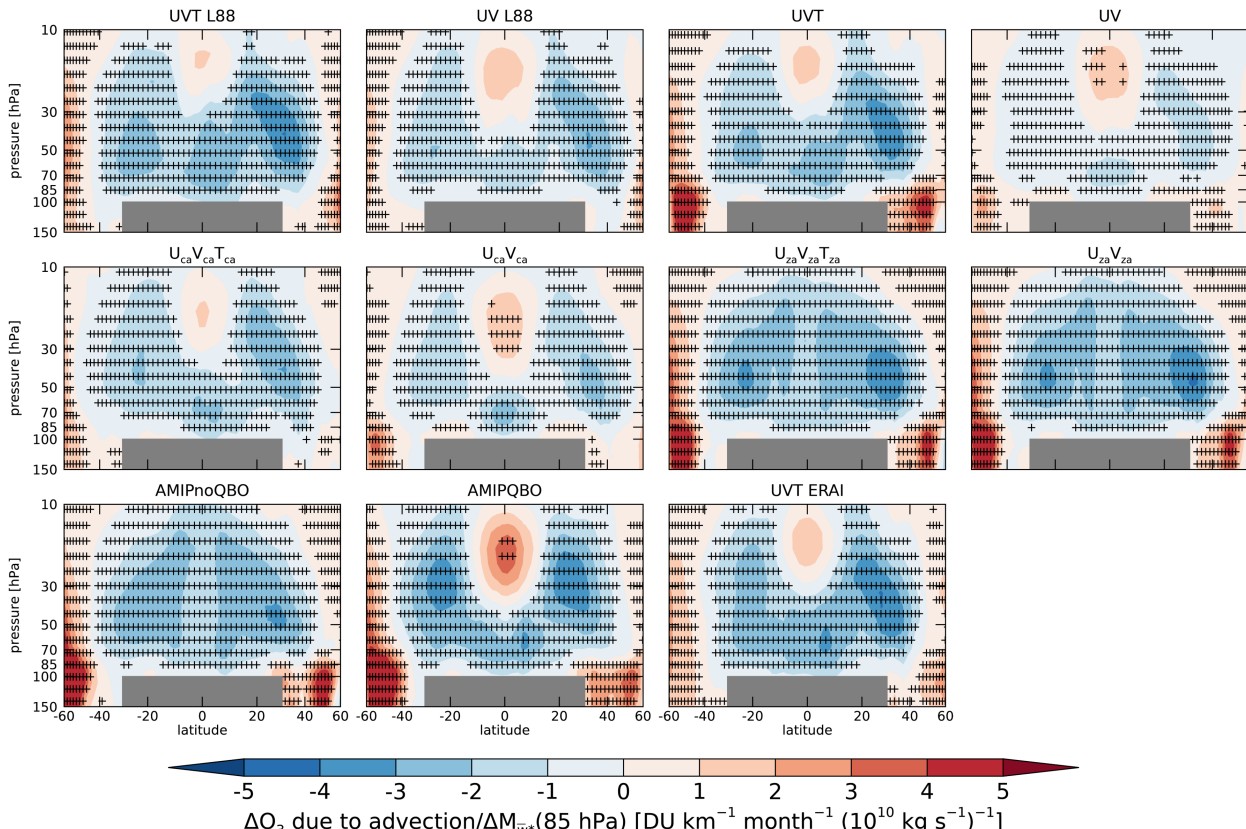

**Figure 7: Regression coefficients from a linear fit of monthly desasonalized anomalies of partial column ozone tendency due to advection against tropical upwelling mass flux anomalies at 85 hPa for the model simulations (1980 – 2016). The horizontal scale is**

**area weighted, so the area and magnitude of the contours below 30 hPa is roughly representative of the LSCO values presented elsewhere. Statistically significant regression coefficients are denoted by plus symbols.**

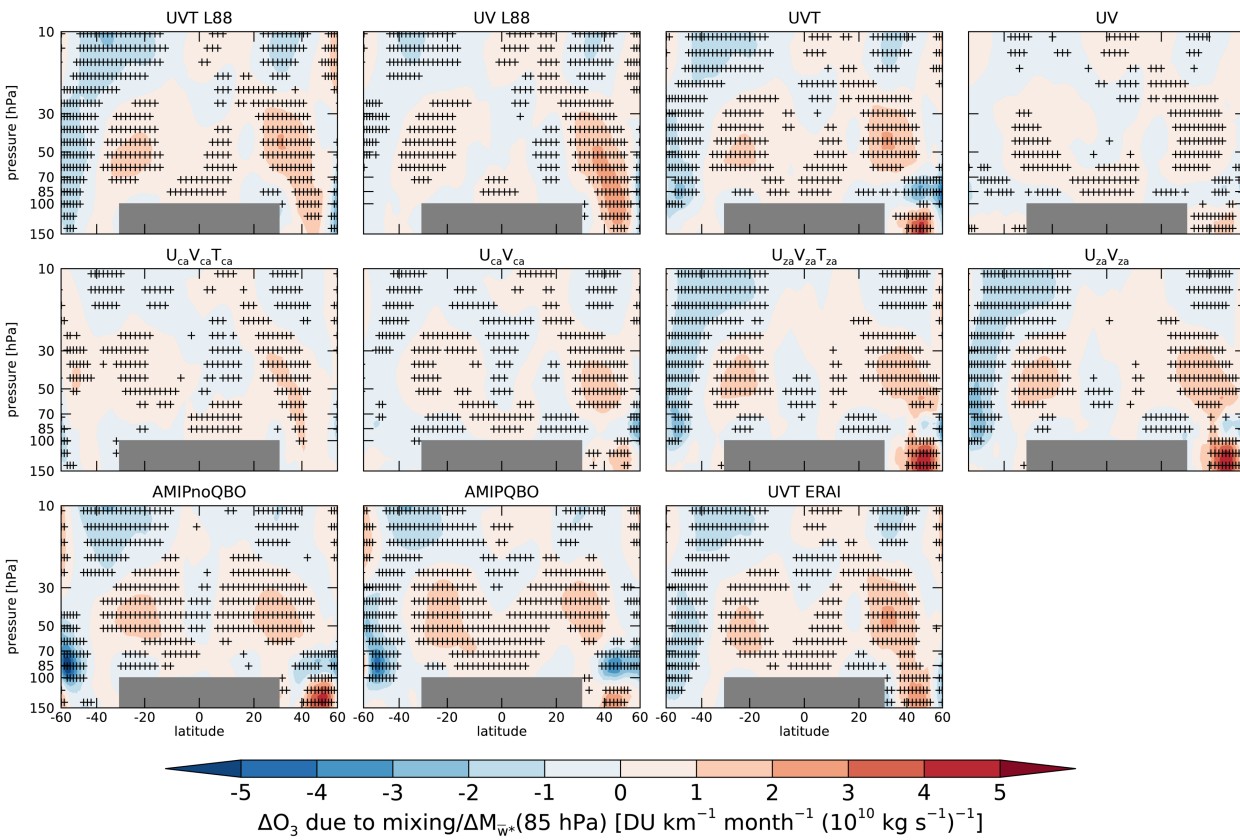

**Figure 8: As in Fig. 7, but for the ozone tendency due to mixing.**

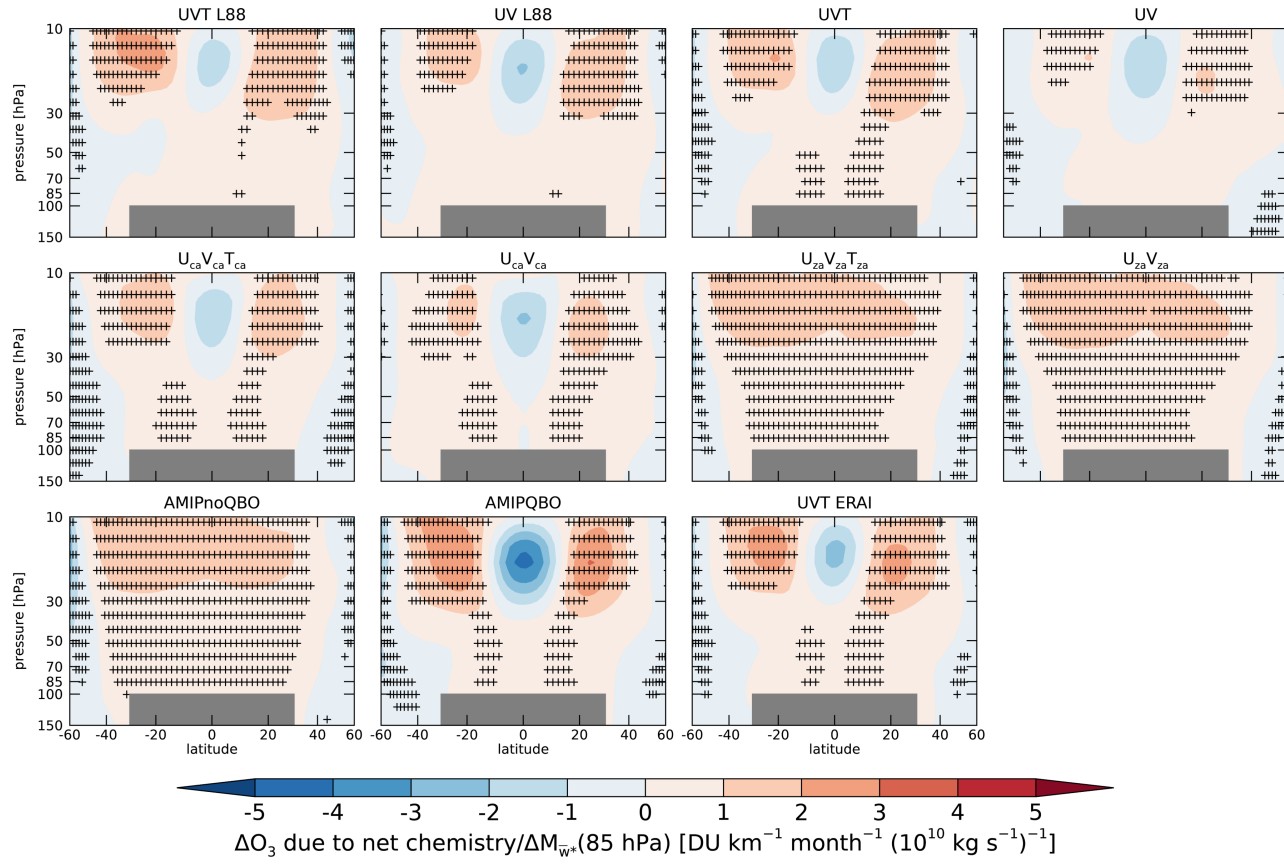

**Figure 9:** As in Figs. 7 and 8, but for the net chemical tendency.

Comparing Fig. 7 to Fig. 5, it is apparent that the ozone advective tendencies are sufficient to explain most of the ozone variability that is congruent with upwelling variability, whereas mixing tendencies do not appear to be as cleanly correlated with upwelling variability. Also, the different patterns of ozone variability in simulations with and without QBO nudging that are present in Fig. 5 are reaffirmed by the ozone advective tendency behavior in Fig. 7. Specifically, simulations without a QBO (i.e., AMIPnoQBO and the zonal anomaly simulations) exhibit a relatively uniform pattern of variability across latitudes and with height. In contrast, simulations with a nudged QBO show a lobe of positive values along the equator above ~30 hPa that suggest anomalous downwelling and downward advection of air with greater ozone mixing ratio (since the vertical gradient of ozone is positive throughout this region). This is because for simulations with a QBO, wind shear anomalies and the anomalous vertical velocities they induce at these upper levels are opposite to those occurring near 85 hPa. Similarly, there

is a clear difference between simulations with and without a QBO in the chemical tendency. Simulations without a QBO show a rather uniform latitudinal pattern of positive anomalies associated with increased upwelling, whereas simulations with a QBO show a negative equatorial lobe above ~30 hPa. This negative lobe is collocated with the positive lobe in the advective tendency. The explanation for this negative lobe in the chemical tendency is that increased advective flux of ozone increases

ozone loss processes (which are proportional to ozone concentration, see Figs. S8-S9).

## 4 Discussion

The claim by B18 of a continuous decline in lower stratospheric ozone since the late 1990's motivated a flurry of research addressing both the robustness of the observational evidence as well as possible dynamical and chemical causes of the purported changes. Regarding the observational evidence, studies have generally confirmed the presence of negative lower

stratospheric ozone trends in the tropics since around 2000 using multiple independent data sources, but the statistical significance of the derived trends is evidently somewhat dependent on methodological details such as start/end dates, latitude/altitude averaging, choice of regressors, and statistical method (e.g., compare DLM-based results here and in Ball et al. 2019 with multiple linear regression technique used in Godin-Beekman et al. 2022). Independent of the issue of observational robustness or statistical significance, several studies have shown convincingly that a pattern of lower

stratospheric ozone decreases is likely to be dynamical in origin, as opposed to being due to novel chemical pathways (e.g., ozone depletion due to very short-lived halocarbons). Even so, the degree to which the dynamical changes relevant to ozone variability are due to externally forced climate change versus internal variability over the previous two decades is still an open question.

Here, we have addressed an inconsistency highlighted by B18, namely that specified dynamics simulations from WACCM not

only fail to produce the observed negative lower stratospheric ozone trends, but to the contrary actually produce significant positive trends in ozone in the recent decades. In this paper, we revisited the WACCM-SD simulation used in B18 (UVT L88) to demonstrate that the model configuration fails to reproduce the underlying tropical upwelling changes present in the MERRA2 reanalysis fields used to drive the model. In contrast to MERRA2's insignificant tropical upwelling trend, the

WACCM-SD simulation used in B18 contains a significant negative tropical upwelling trend around 85 hPa over 1998 – 2016, the time period during which B18 identified negative LS ozone trends in satellite observations. This motivated an exploration of whether the negative upwelling trend in the B18/UVT L88 simulation could account for the apparent discrepancy between modeled and observed ozone trends.

Additionally, we also considered a suite of free-running and specified dynamics simulations with alternative nudging configurations in addition to the standard UVT L88 configuration, as large variability in their upwelling has been previously recognized (Davis et al., 2020). We demonstrated a wide spread in short term upwelling trends amongst the suite of simulations, as well as within three modern reanalyses. Along with this large spread in upwelling variability and trends, we also showed a large spread in short-term ozone trends, with some simulations showing large negative trends in the lower stratosphere and others showing large positive trends.

To interpret these results, we identified a strong relationship between monthly anomalies of lower stratospheric tropical upwelling and near-global lower stratospheric ozone column within each simulation. Given the strong sensitivity of ozone to upwelling at 85 hPa, we then considered the vertical/latitudinal pattern of co-variation between ozone and tropical upwelling at 85 hPa. We identified a "bird-shaped" pattern of ozone variability throughout the stratosphere for simulations that included a (nudged) QBO, in contrast to simulations lacking a QBO (Fig. 5). This pattern emerges in the observations (Fig. 5, SWOOSH/MERRA2), suggesting that the pattern may be a useful diagnostic with which to evaluate ozone fields in chemistry-climate models containing a spontaneously-generated QBO. Finally, based on an ozone budget analysis, we demonstrated that, perhaps unsurprisingly, advection is the dominant process through which ozone variations are driven by variability in lower stratospheric upwelling.

In addition to the monthly ozone-upwelling relationships identified in the simulations and observations, we also identified a relationship across simulations for the trend in tropical upwelling at 85 hPa and trend in near-global lower stratospheric ozone column. Put simply, short-term ozone trends scale linearly with short-term trends in tropical upwelling. Model simulations that increase tropical upwelling have more negative ozone trends throughout a broad range of the stratosphere, and vice versa for simulations with negative upwelling trends.

As we demonstrated, none of the WACCM simulations capture the observed ozone decline present in the SWOOSH or BASIC-SG data sets over the 1998-2016 time period. However, the ozone/upwelling scaling from WACCM (Fig. 6) suggests that roughly half of the difference between the observed and modeled LSCO trends in B18 is explained by the spurious upwelling trend in the UVT L88 simulation. The other half of the difference remains unexplained, and overall, the analysis presented here suggests a remaining inconsistency between WACCM simulations and observations. As an example, the ozone/upwelling scaling in the suite of WACCM simulations suggests that a large short-term upwelling trend of $\sim6\%$ decade$^{-1}$ would be needed to explain the satellite trends over the 1998 to 2016 time period, which is consistent with the ERAI trend but not with trends from the ERA5 and MERRA reanalyses. Alternatively, if one takes at face value the ERA5/MERRA2 upwelling trends and WACCM ozone/upwelling scaling, then the observational trend estimates of near-global lower stratospheric ozone column are too negative.

In principle, the strong relationship between lower stratospheric partial column ozone and 85 hPa upwelling mass flux should provide for a solid observational constraint on model behavior. However, the large range of reanalysis upwelling estimates and inability of the SD simulations to reproduce the upwelling from their input reanalyses severely limits the utility of SD simulations for accurately reproducing recent ozone variability. Somewhat ironically, a less constrained free-running simulation using only surface boundary conditions and a nudged quasi-biennial oscillation (QBO) more closely captures both interannual variability and decadal-scale ozone "trends" than the nudged model simulations. As previously recognized (Davis et al., 2020), nudged model simulations fail to constrain gravity wave momentum forcing, which can contribute significantly to upwelling. Future studies aimed at better representing interannual to decadal variability in gravity wave momentum forcing in specified dynamics simulations may thus have the potential to help better quantify lower stratospheric ozone variability on these timescales.

**Code/Data availability**

460    SWOOSH data are publicly available at https://csl.noaa.gov/swoosh. Output from the WACCM simulations is available on request to the authors. ERA-Interim and ERA-5 reanalysis data are freely available from the European Centre for Medium-Range Weather Forecasts (https://apps.ecmwf.int/datasets/data/interim-full-moda/, European Centre for Medium-Range Weather Forecasts (ECMWF); https://cds.climate.copernicus.eu/#!/search? text=ERA5&type=dataset, Copernicus Climate Change Service (C3S)). MERRA-2 reanalysis data are freely available from NASA

465    (https://doi.org/10.5067/WWQSXQ8IVFW8, Global Modeling and Assimilation Office).

**Author contributions**

SD lead the analysis and writing. ND and RP performed model simulations and contributed to the analysis and writing. ER and KR contributed to the analysis and writing.

**Competing interests**

470    The authors declare no competing interests.

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
