# Peer review of "The role of tropical upwelling in explaining discrepancies between recent modeled and observed lower stratospheric ozone trends"

_EGUsphere, 2022_

## Author Comment (AC1)

"The role of tropical upwelling in explaining discrepancies between recent modeled and observed lower stratospheric ozone trends", ACPD, 2022, by Davis, S. M. et al..

In their paper, Davis et al. analyse a suite of WACCM model experiments with, without, and with different sorts of nudging as to whether the simulations can reproduce observed ozone trends from the Ball et al. (2018) study. They find that a misrepresented trend in tropical upwelling in the original simulation setup leads to some of the ozone discrepancies. Particular nudging setups help to get closer to observations with some regard, but surprisingly, the free-running simulation (with nudged QBO) represents the ozone trends best. Overall, I think this is an interesting and well-written paper, the method and the analysis are elaborate and the study reveals some new insights to this topic, which has shaken up the community for several years now. I would be happy to see the paper published in ACP soon. I do, however, have a few remarks that I think are important to consider before publication and two of these, I guess, should be viewed as major points. Revising the paper in that sense should not be too cumbersome, though, please see below.

**Thank you for the encouraging and thoughtful review, Roland! Please see our responses below in bold.**

Major issues:

• Sect. 3.4: The discussion about mixing appears oversimplified to me. What type of mixing is it that you diagnose? Is it parameterised horizontal/vertical diffusion or is it Kyy or some of such diagnostics (see e.g. Abalos et al. (2016, 10.1175/JAS-D-16-0167.1) or Eichinger et al. (2019, 10.5194/acp-19-921-2019))? Moreover, did you consider the diffusivity of the model's advection scheme? I assume this is still included in what you call 'advection'. If that is the case, the results may be somewhat misinterpreted. See Dietmüller et al. (2017, 10.5194/acp-17-7703-2017) for details and for some quantification of this effect. Moreover, please consider e.g. Eichinger et al. (2019, 10.5194/acp-19-921-2019) for the discussion of the impact of mixing on tracer trends and Dietmüller et al. (2021, 10.5194/acp-21-6811-2021), Orbe et al. (2020, 10.1029/2019JD031631), Ball et al. (2020, 10.5194/acp-20-9737-2020) for discussing the influence of mixing on ozone in the extratropical lower stratosphere. Revise also L390-391 accordingly.

**The "mixing" discussion in section 3.5 (formerly section 3.4) is referring to the ozone tendency due to mixing from the TEM formalism, as discussed briefly in Section 2 (lines 115-118 of the original manuscript). Our ozone tendency due to mixing is essentially identical to that from the Abalos et al 2016 paper referenced above.**

**Looking back at this, we agree that the discussion on mixing in this section is a bit lacking. In fact, there was a reference in the former section 3.4 to an equation 1 that did not exist (but was in an earlier draft of the manuscript). To address this, we've added back in that text in section 2, including the TEM tracer budget equation, to make clear that we are considering the ozone tendency due to resolved to mixing in our analysis.**

• My other concern (which is partly linked to the above) is the averaging over 60S-60N. It has been shown in several studies, and the present study shows it again, that different processes are at work in the tropics and in the extratropics with regard to LS ozone trends and variability, see for example Dietmüller et al. (2021, 10.5194/acp-21-6811-2021), Orbe et al. (2020, 10.1029/2019JD031631), Ball et al. (2020, 10.5194/acp-20-9737-2020). The tropics are controlled by dynamical upwelling within the pipe, in the extratropics, mixing is important, but extratropical downwelling and possibly other processes can have an impact as well. I understand that the trends from the Ball et al. (2018) study are meant to be reconciled, but in light of the studies above, I think doing it for 60S-60N is prone to be misleading in interpretation. The relevant figures (mainly Figs. 4 and 6) should be shown for tropics (as in supplement) and extratropics separately and the results should be discussed with respect to the different regions then as well. Also the introduction completely lacks these points, which I think are very relevant for the topic. In Sect. 3.3 this point is already taken up, but only very (too) briefly. I assume that when doing the separation, more than 'roughly half' (see L298) can be explained in the tropics, and it will be very interesting to see how much can be explained in the extratropics.

**The reviewer is correct that we used the 60°S – 60°N latitude band in order to replicate the Ball et al. 2018 methodology, and agree that different processes such as mixing can be more dominant in certain regions such as the extratropics. However, we would like to point out that in our analysis, we are not using upwelling and vertical advection of ozone interchangeably. While we are largely focused on the strong correlation between upwelling and ozone, we are emphatically not claiming that ozone advection due to upwelling is the sole process responsible for ozone variability. Rather, we are pointing out that all of the processes that control ozone (including both mixing and chemistry) covary with upwelling, and not just advection. We have attempted to make this point clearer in the manuscript.**

**To the reviewer's specific point about considering different latitude bands, we note that we already included a version of Figure 4 over the tropics (30°S to 30°N) as Fig. S1 in our first submission. Also, Figure 5 shows the relationship between ozone and upwelling variability as a function of latitude to demonstrate that the relationship in Figure 4 is not simply confined to the tropics. In addition to these existing figures, we've also added supplemental versions of Figure 4 (Figs. S2 and S3) and Figure 6 for the tropics (Fig. S5), NH extratropics (Fig. S6), and SH extratropics (Fig. S7). We have added discussion in the manuscript to support these figures.**

Minor issues:

• L13: Add (short or half) sentence that tropical upwelling is important for ozone in LS.

**Thank you for pointing this out. We agree the transition here was abrupt and have re-worded the abstract.**

• L15: state more precisely the time span you are analysing

**Done**

• L24: Due to the upwelling trends again, or other reasons too? Maybe state briefly here.
**Done. We've re-worded this sentence to better contrast the free-running model results to the SD results in the previous sentence.**

• L27: ... Montreal Protocol in 1987, the atmospheric concentration of ozone depleting substances (ODSs) is declining, and ...

**Done**

• L55: "replay" simulations. Is that something people know about, I am afraid I do not. Can you explain what it means?

**"Replay" simulations are similar to nudging used by the WACCM-SD simulations described in this paper, but the methodology is different. In addition to the existing Wargan reference in this sentence, we've added a reference to the Orbe et al 2017 paper that describes the replay methodology as well as the ability of these simulations to reproduce important tracer transport features.**

**Orbe, C., Oman, L. D., Strahan, S. E., Waugh, D. W., Pawson, S., Takacs, L. L. and Molod, A. M.: Large-Scale Atmospheric Transport in GEOS Replay Simulations, Journal of Advances in Modeling Earth Systems, 9(7), 2545–2560, doi:10.1002/2017MS001053, 2017.**

• L74: The terms 'nudging timescale' and 'meteorology frequency' are not self-explanatory, can you modify the wording or explain it more precisely please.

**We've provided the definition of nudging timescale, "(i.e., the timescale over which the model field is relaxed towards the reanalysis fields)", and have changed "meteorology frequency" to "reanalysis data frequency (e.g., 3-hourly versus 6-hourly)" to explain the distinction between the two.**

• L80: Would there be a citation for this paradoxicality? If not, can you explain how this comes about?
**Yes. This is discussed in Davis et al., 2022. We've added a reference here.**

• Section 2: Partly for the sake of reproducibility, partly for transparency, can you add some information on the following:

**At the beginning of section 2, we refer to the Davis et al., 2020 paper, which comprehensively describes the simulations we are analyzing, and our description is thus**

**brief. However, we have added the requested information as described in our responses below**

– There are some nudging parameters, such as nudging strength, can you provide it, do you vary it, is this a standard setting, have you considered varying it, is there history on that?

**We added the nudging timescale here, which is the "standard" configuration for CESM. It is also discussed in further detail (in addition to sensitivity testing of the timescale) in Davis et al., 2020, which we reference.**

– What is the approximate vertical resolution in the UTLS in your setups

**The WACCM L66 grid is virtually identical to MERRA2 between 72 hPa and 266 hPa (inclusive), and has a vertical resolution of 1 km in this region.**

– Do you nudge in all altitudes, i.e. in all levels? If not, where not?

**Nudging occurs uniformly at all levels below 0.8 hPa, and the strength of nudging is linearly reduced from 1% per time step to 0% between 0.8 hPa and 0.2 hPa. We've added this information in the text.**

– Is nudging performed in grid-point space or in spectral space in WACCM?

**Nudging is performed in grid-point space. This is now mentioned in the text.**

– What SSTs and SICs do you use?

**We use the default data ocean model in CESM, which comes from Hurrell et al 2008. We added text and reference to this paper.**

**Hurrell, J. W., Hack, J. J., Shea, D., Caron, J. M. and Rosinski, J.: A New Sea Surface Temperature and Sea Ice Boundary Dataset for the Community Atmosphere Model, J Climate, 21(19), 5145–5153, doi:10.1175/2008JCLI2292.1, 2008.**

• L105: I understand now that with 'native' levels, you mean the ERA-I levels, that was not clear to me when I first read it, please clarify. Do you conduct this simulation simply because it is technically possible, and you thought this might have an impact, or is there some other rationale behind this experiment?

**No, the use of the term 'native' in this sentence refers to WACCM's 'native' levels (the L66 grid), not that of ERA-I. We've modified this sentence to make this clearer. The rationale for doing SD simulations on the L66 grid is explained in Davis et al. 2020, but we realized that we didn't summarize that rationale in this paper, so we've added the following:**

*"In addition to simulations with the default SD configuration, we also use 66-level SD simulations as in Davis et al. (2020) in which MERRA-2 has been interpolated to WACCM's native vertical grid. The motivation for these simulations is to reduce spurious latent heating and gravity wave momentum forcing due to parameterizations for convection and gravity wave generation, respectively. These parameterizations respond dramatically differently when used with the L88 vertical grid, as the L88 grid contains approximately triple the number of vertical levels in the boundary layer and lower troposphere."*

• Sect. 2: I think you'll have to argue why you didn't perform a simulation nudged to ERA-5. ERA-Interim is outdated, I think also the (tropical upwelling) trends in ERA-5 are different now. This simulation is clearly missing. Maybe you can argue that MERRA is assimilated in ERA-5, but I am not sure if that is enough, or do you want to keep it for a follow-up study? If a simulation of that kind should be available (or doable on the quick), include it please, if not, at least discuss the possible differences when using ERA-5 for nudging.

**We agree that considering the newest reanalysis products when analyzing upwelling is worthwhile, and note that we've already included ERA5 in our analyses. Specifically, it is included in Figures 1, 2, and 6, with the ERA5 trend results shown in Figure 2b. These figures (particularly Figure 2) illustrate that ERA5 is highly correlated with ERA-I above 85 hPa and (like MERRA-2) does not show a significant trend at 85 hPa over 1998-2016. The point here isn't whether MERRA-2 or ECMWF reanalyses are right, it's that nudging to a reanalysis doesn't necessarily give you back the same dynamical trends as the input. Because of this and our previous work in both Davis et al. 2020 and Davis et al. 2022, we did not do an ERA5 SD simulation and do not believe it will add further clarity to the paper or impact the conclusions in any significant manner.**

• L111: Usually simulations like that are stuck in one QBO phase, can you state in which one it is here?

**That is correct. This version of WACCM has persistent easterlies in the equatorial stratosphere. We have noted this in the manuscript.**

• L113-125: Can you be more precise about the diagnostic calculations here. I understand that you calculate w*-based upwelling from the model and v*-based tropical upwelling from the reanalyses, but I don't understand why you would do that. I think this should be done as consistent as possible.

**We thank the reviewer for this comment. The text original description was incorrect on using w\* vs. v\*. We used v\* for both calculations, although we also did sensitivity testing using both that wasn't shown in the manuscript. The text has been updated.**

Moreover, for usage of w\* please consider our paper Eichinger and Sacha (2020, https://doi.org/10.1002/qj.3876), such that density is correctly chosen for upwelling calculation for the particular w\*, this can have an impact on the upwelling trends.

**We acknowledge the issue brought up by this paper, which relates to the conversion from omega units (Pa s⁻¹) to vertical velocity units (m s⁻¹) using a fixed scale height. However, this issue does not apply to our analysis because we do not analyze w\* in this paper. Rather, our analysis uses upwelling mass flux (units kg s⁻¹) based on the TEM streamfunction.**

• L146: It is here a little unclear why you use 85 hPa, the standard in literature is 70 or 100 hPa. Later on it makes sense, as you have the best correlation there, but here this choice requires an explanation. Or was this level chosen in previous papers already (and I missed it)? If so, please state it.

**We've added the following sentence:** *"The 85 hPa level is the first level above the climatological tropical tropopause in WACCM, and, as will be shown later in the paper, is the level most closely associated with ozone variability."*

• L231: State why you would expect that.

**We modified this to read (see underlined) :** *"As might be expected if anomalous increases in upwelling lead to increased advection of ozone-poor air into the stratosphere, ..."*

• Sect. 3.3: I was sceptical at first, when I read that you want to conclude from variability to trends, as the processes that control variability (ENSO, QBO, seasonal cycle, ...) are different from those that control decadal trends (GHG and ODS emissions,...), however, I think the way it is conducted in this paper is absolutely fine. What I would still ask you to do is to write a few words about this, and why you think you can actually conclude from variability to trends in your case, mainly to encourage readers to think about this before doing it.

**We agree that the segue between variability and trends could be improved. We thus created separate sub-sections for variability (now Sect 3.3) and trends (new Sect 3.4). The step from variability to trends is addressed at the beginning of the new Sect 3.4:**

*"Given the correlation between upwelling and LSCO for deseasonalized monthly anomalies, we next turn our attention to whether or not the multi-decade trends in these quantities are related. While the processes that control seasonal to interannual variability (e.g., ENSO, QBO) are expected to be different than those controlling decadal trends (e.g., GHG and ODS emissions), it is possible that similar relationships between upwelling and LSCO exist across*

*timescales due to the modulation of upwelling by a variety of processes."*

• Discussion around Fig. 4: Maybe it could help to mention the influence of chemistry (and dynamics) when it comes to explaining the low correlations above around 50 hPa.

**We added text mentioning the decreased chemical lifetime at higher altitudes.**

• Sect. 3.4 Could you additionally show the impact of chemistry on ozone trends (maybe in appendix or supplement). Would these 3 parts then (in the stratosphere) add up to the total tendencies or what else would still be missing?

**Good idea. We added the plot for the net chemical ozone tendency as Figure 9, and have added some discussion of that in Sect. 3.4. The total ozone tendency is indeed the sum of the advective, mixing, and chemical tendency terms in the TEM framework. However, the TEM framework is an approximation for the true tracer tendency. As discussed in Abalos et al 2013, there is a residual term, and this residual term is large enough that it is not possible to attribute relatively small ozone changes in the model using the tendency terms.**

• L381: What bird-shaped pattern? You never mentioned this before, where was that?

**Good catch, our mistake! The "bird-shaped" pattern can be seen for the QBO simulations in Figure 5. We've added text describing this pattern in section 3.3 where we discuss Figure 5.**

Technical issues:

• L13: ....that, despite the nudging, the ...

**The abstract has been re-worded here.**

• L17: ...lower stratospheric ozone

**Done**

• L45: ...variability that strongly determines ozone variability ...

**Done**

• L60: ...stratospheric ozone decline...

**Done**

• L145: runs → simulations (and many other times)
**Done**
• L162: don't → do not
**Done**

• L182: remove 'also'
**Done**

• L199: Remove (or replace) 'Interestingly' (everything you write is interesting, right!)
**Done**

• L200 and in many other places: To my understanding, the lowermost stratosphere is the part in the extratropics that is at around the pressure altitude as the tropical upper troposphere. So in the tropics there is no lowermost stratosphere, but rather just the 'tropical lower stratosphere'. Please check if I am mistaken, if not, change it throughout the paper.

**Done**

• L210: State what latitudes you are talking about here.

**Done**

• L230: State what latitudes you are talking about here.

**Done**

• L257: include 'in the lower stratosphere'

**Done**

• L260 remove 'are'

**Done**

• L301: remove 'very'

**Done**

• L314: don't → do not

**Done**

• L344-348: Split sentence into 2 or 3 sentences.

**Done**

• L352: Remove first 'the'

**Done**

• L368: Remove 'highly'
**Done**

• L376: variability → trends and variability

**Done**

• L395: remains unexplained, or, is still unexplained

**Done**

• L418: Something seems wrong here, 'dlmmc'.

'dlmmc' is the name of the DLM software package, and is part of the title of the publication.

---

## Author Comment (AC2)

This paper investigates changes in lower stratospheric ozone over the recent past in model simulations and satellite observations, an important topic when assessing the effect of the Montreal protocol or effects of climate change on stratospheric dynamics. It is found that lower stratospheric ozone changes scale linearly with tropical upwelling velocity across different model simulations with different nudgings, suggesting a primary role of tropical upwelling for controlling decadal ozone changes in that region. However, none of the model simulations reproduces the observed ozone trend and, if the linear ozone-upwelling relation holds, a large upwelling trend would be needed to explain the observed ozone trend. Furthermore, nudging the model dynamics towards reanalysis turns out to be tricky, such that tropical upwelling trends in nudged simulations are often very different from the original reanalysis trends, such that the usefulness of nudged simulations to investigate observed ozone variability appears questionable.

The topic of lower stratospheric ozone trends is of much interest to the stratospheric community, and this paper makes an important contribution to further our understanding on these trends. The paper is concise, well structured and well written and the results are presented clearly, and I enjoyed reading. I do strongly recommend publication and only have a few minor and specific comments, which could help to further improve the presentation and discussion.

**Thank you for the review! We have addressed the minor comments below in bold.**

**Minor comment:**

I don't fully agree with the statement that ERA-Interim shows "inconsistency of its long-term upwelling trend against ... observations..." (L315), or "ERA-Interim being a particular outlier" (L375). Indeed, there is an inconsistency, but only if one considers residual circulation upwelling velocity calculated using the standard TEM residual circulation definition, and also only for a particular period (e.g. Seviour et al., 2011). On the other hand, the ERA-Interim upwelling calculated from momentum or thermodynamic balances shows a long-term increase in the lower stratosphere (Abalos et al., 2015) which is (at least qualitatively) consistent with observational estimates (e.g., Ray et al., 2014). Also, mean age trends based on upwelling from thermodynamic balance estimate (diabatic heating rates) from ERA-Interim appear to agree better with mean age observations than other reanalyses (e.g., Ploeger et al., 2019, 2021). I'd suggest to include a more careful discussion (e.g. L315ff) and also reconsider the statement that "upwelling trends explain roughly half of the discrepancy between modeled and observed ozone changes" (e.g., L297ff, L395), as this is related to the former one. There are a few more specific comments below related to that.

**We appreciate the reviewer's comment on this, and have updated the text to include the information provided by the reviewer, including discussion of the fact that ERA-I's "outlier" status is related to the standard TEM definition (near line 315 in the original text). We have noted that the ERA-Interim upwelling trend (coupled with WACCM's upwelling-LSCO trend relationship) is consistent with observations, and have removed the value judgements regarding ERA-Interim.**

**Regarding upwelling trends explaining half of the discrepancy, we have addressed this issue in the specific comments below.**

**Specific comments:**

L166: Remarkably, the QBO-related upwelling increase during 2015-2016 in AMIPQBO, which is likely responsible for the positive upwelling trend (at least partly), is not seen in the original reanalysis data. This could be worth a note.

**We added a note on this.**

L180, Fig. 2: Why is the correlation between the SD-simulations and their respective reanalysis decreasing below about 70hPa? Is the nudging strength varying with level?

**We don't have a good explanation for this decrease in correlation below 70 hPa. The nudging strength is uniform below 0.8 hPa, which we have noted in Section 2 (also in response to RC1). We can speculate here that differences in the TTL structure between the model and reanalyses may lead to inconsistencies in the conversion of wave variability to upwelling.**

L201, Fig. 2: Any idea why the nudging of the climatology shifts trends to be less negative / more positive in the lower stratosphere (i.e. nudging only anomalies results in more positive upwelling trends)? The same happens for nudging T, in particular for the "ca" (green) case.

**We speculate that nudging absolute values of winds and temperatures (as opposed to anomalies) can act as (potentially unwanted) diabatic heating and impact gravity wave momentum forcing, as discussed in Davis et al. 2020. We have added a note on this in Section 2 where we motivate the alternative SD configurations.**

Fig. 2 and 3: Related to the last comment, I'd find it noteworthy that nudging temperature climatological anomalies (T-ca) changes the lower stratospheric upwelling trend from negative to positive and the lower stratospheric ozone trend from positive to negative. For the zonal anomaly nudging simulations this is not the case.

**We agree that this is noteworthy, and have added a few sentences mentioning this.**

Fig. 3: Another interesting detail is that the extent of negative ozone trends into the NH middle latitudes is not reproduced by any simulation. This might point to mixing effects which are perhaps not well represented in the model (as also suggested by Wargan et al., 2018; Orbe et al., 2020). Maybe also worth mentioning.

**Agreed. Done.**

L239, Fig 4: I think it could make sense to include the figures for tropical latitudes also in Fig. 4, just to show how clear the relation is for the region where we expect it to be clearest.

**The tropical latitude version of this was already included as Fig. S1.**

L297: I don't understand this remark ("...the negative trend in upwelling in that simulation appears to explain roughly half of..."). My problem is that we don't know the true upwelling. If ERA-Interim would be the truth (and not MERRA-2) its positive upwelling trend would be in the range where the linear relation in Fig. 6 is consistent with the observed ozone trend.,

so that the entire ozone trend difference could be explained by the upwelling trend difference. (This is related to my minor comment above).

**The key point in this statement is that for these simulations MERRA-2 is the input to the reanalysis, so it is the "truth" to which we are comparing (even though we of course don't know the true upwelling). The sentence after this one clarifies that the upwelling-LCSO relationship in Fig. 6 can account for around half of the difference in LSCO trend between the UVT L88 simulation (~ +1.2 DU/decade) and SWOOSH (~ -1 DU/decade). We've modified the second sentence to try and make this clearer.**

L375: Only trends in ERA-Interim upwelling calculated using the standard definition of TEM residual circulation velocities are an outlier (Abalos et al., 2015). (This is related to my minor comment above).

**We've removed the reference to ERA-Interim being an outlier here.**

L394: Also here it is not entirely clear to me what is exactly meant. Here, I understand that 50% of the trend difference can be attributed to the spurious upwelling trend due to nudging - and with this statement I would agree. Above (L297), it was not so clear to me what was meant. (Also related to my minor comment above).

**We don't understand what the reviewer doesn't understand here. The reviewer is interpreting this line correctly, i.e., "that 50% of the trend difference can be attributed to the spurious upwelling trend due to nudging".**

L395: Given the linear relation in Fig. 6, isn't it most likely that the simulations underestimate the true upwelling trend? If the true upwelling trend would be positive - similar to ERA-Interim - this would explain the difference. Couldn't this be hypothesized here? (Also related to my minor comment above).

**We don't know what is most likely, but the model upwelling-LCSO trend relationship suggests that the negative ozone trends in observations are consistent with a positive upwelling trend. We've changed the text here to note that the Fig. 6 relationship is consistent with the ERAI upwelling trend and observations.**

**Technical corrections:**

L325: I can't find eqn. 1.

**This was added back in. See also response to RC1**

L352: There is one "the" too much.

**Fixed**